# Short-chain fatty acids activate acetyltransferase p300

Sydney P Thomas[1,2], John M Denu[1,2,3]*

[1]Wisconsin Institute for Discovery, Madison, United States; [2]Department of Biomolecular Chemistry, University of Wisconsin – Madison, Madison, United States; [3]Morgridge Institute for Research, Madison, United States

**Abstract** Short-chain fatty acids (SCFAs) acetate, propionate, and butyrate are produced in large quantities by the gut microbiome and contribute to a wide array of physiological processes. While the underlying mechanisms are largely unknown, many effects of SCFAs have been traced to changes in the cell's epigenetic state. Here, we systematically investigate how SCFAs alter the epigenome. Using quantitative proteomics of histone modification states, we identified rapid and sustained increases in histone acetylation after the addition of butyrate or propionate, but not acetate. While decades of prior observations would suggest that hyperacetylation induced by SCFAs are due to inhibition of histone deacetylases (HDACs), we found that propionate and butyrate instead activate the acetyltransferase p300. Propionate and butyrate are rapidly converted to the corresponding acyl-CoAs which are then used by p300 to catalyze auto-acylation of the autoinhibitory loop, activating the enzyme for histone/protein acetylation. This data challenges the long-held belief that SCFAs mainly regulate chromatin by inhibiting HDACs, and instead reveals a previously unknown mechanism of HAT activation that can explain how an influx of low levels of SCFAs alters global chromatin states.

## Editor's evaluation

This study investigates the mechanism of agents like butyrate as metabolites that affect histone acetylation. The authors make the unexpected and interesting finding that such metabolites can stimulate the activity of the acetyltransferase p300 rather than the commonly accepted concept that they block histone deacetylases. The authors show evidence that p300 stimulation involves acylation of Lys residues on its autoinhibitory loop. The authors have effectively responded to prior concerns raised by the reviewers. This study should be of broad interest to the epigenetic research community.

*For correspondence:
john.denu@wisc.edu

Competing interest: The authors declare that no competing interests exist.

## Introduction

Short-chain fatty acids (SCFAs) play a crucial role in human health. Although SCFAs include any fatty acid with fewer than six carbons, the first three members of the family—acetate, propionate, and butyrate—are by far the most abundant physiologically (*den Besten et al., 2013*; *Koh et al., 2016*; *Tan et al., 2014*). These three SCFAs are produced in large quantities by bacterial fermentation of nondigestible fiber, and levels can fluctuate based on the amount and type of fiber in the diet (*Bird et al., 2000*; *den Besten et al., 2013*; *Jenkins et al., 1998*; *Levrat et al., 1991*; *Marsono et al., 1993*; *Ríos-Covián et al., 2016*; *Velázquez et al., 2000*). In humans, total SCFA concentrations can reach over 100 mM in the colon, in a ratio of ~60:20:20 acetate:propionate:butyrate (*Cummings et al., 1987*; *Parada Venegas et al., 2019*). SCFAs alone may provide up to 10% of daily caloric requirements in humans (*Bergman, 1990*). Colonocytes are especially prone to metabolize SCFAs, deriving

60–70% of their energy from SCFA oxidation (*Roediger, 1982*). Excess SCFAs are transported to the liver through the portal vein and are eventually released into the peripheral blood, existing at micromolar concentrations (*Bloemen et al., 2009*; *Cummings et al., 1987*).

Butyrate and acetate are the most-studied members of the group. Acetate plays vital roles in cellular metabolism, especially as a building block for the central metabolite acetyl-CoA (*Bose et al., 2019*; *Liu et al., 2018*; *Zhao et al., 2016*). The physiological effects of butyrate are abundant: butyrate has been reported among other things to improve intestinal barrier function (*Chang et al., 2014*; *Hamer et al., 2007*; *Peng et al., 2009*), reduce inflammation (*Chriett et al., 2019*; *Usami et al., 2008*; *Vinolo et al., 2011*), improve metabolic health (*Gao et al., 2009*; *Lin et al., 2012*; *Müller et al., 2019*), and prevent cancer (*Donohoe et al., 2012*; *Hague et al., 1993*). The role of propionate is less clear—while it shows the same beneficial effects as butyrate in many studies, extremely high levels of propionate are associated with negative health outcomes such as propionic acidemia and autism spectrum disorders (*Abdelli et al., 2019*; *Al-Lahham et al., 2010*; *Li et al., 2017*).

It is generally believed that SCFAs induce physiological effects by changing the cell's epigenetic state (*Hamer et al., 2007*; *Hinnebusch et al., 2002*; *Krautkramer et al., 2016*; *Tan et al., 2014*). Epigenetics describes the regulation that takes place in eukaryotic chromatin. Much of this regulation centers around histone proteins, which are heavily decorated with post-translational modifications (PTMs). Histone PTMs affect a wide array of processes, from DNA accessibility to transcription factor binding (*Bannister and Kouzarides, 2011*; *Karch et al., 2013*; *Kouzarides, 2007*). Histone PTMs include lysine acetylation, which opens chromatin and makes it more accessible to transcription, and lysine methylation, which performs various functions. Recently, a host of other histone PTMs have also been identified, including lysine propionylation and butyrylation (*Chen et al., 2007*; *Dai et al., 2014*; *Goudarzi et al., 2016*; *Kebede et al., 2017*).

Histone PTMs, as well as the enzymes which add or remove them, are exquisitely sensitive to changes in cellular metabolism (*Albaugh et al., 2011*; *Fan et al., 2015*; *Sassone-Corsi, 2013*). Several central metabolites are also substrates for histone modifications: such as acetyl-CoA, which is used to acetylate histones. Histone acetylation is regulated on two fronts—first by histone acetyltransferases (HATs), which transfer acetyl groups to chromatin, and then by histone deacetylases (HDACs), which remove them (*Albaugh et al., 2011*). Acetate, propionate, and butyrate have all been reported to inhibit HDACs to varying extents, with butyrate being the most potent inhibitor (*Bolduc et al., 2017*; *Candido et al., 1978*; *Davie, 2003*; *Hsu et al., 2016*; *Silva et al., 2018*; *Waldecker et al., 2008*). HDAC inhibition has been proposed as a mechanism for SCFA's anti-cancer effects (*Donohoe et al., 2014*; *Hinnebusch et al., 2002*), insulin regulation (*Chriett et al., 2019*; *Gao et al., 2009*), and immunomodulation (*Bolduc et al., 2017*; *Chang et al., 2014*; *Usami et al., 2008*; *Vinolo et al., 2011*). However, there are still questions as to whether SCFAs regulate the epigenome by HDAC inhibition alone (*Corfe, 2012*; *Donohoe et al., 2012*; *Gibson, 2000*).

It has been recently reported that the gut microbiome can affect histone modifications systemically (*Krautkramer et al., 2016*; *Kumar et al., 2014*; *Romano et al., 2017*; *Takahashi et al., 2006*; *Wellen et al., 2009*; *Zhao et al., 2020a*). Our lab has demonstrated that germ-free mice display distinct patterns of histone PTMs in multiple tissues (*Krautkramer et al., 2016*). This phenotype was complex, but could be generally characterized by decreased histone acetylation on multiple sites. However, acetylation levels did not depend solely on bacteria—colonized mice fed diets that produced low levels of SCFAs showed similar PTMs to germ-free mice. Finally, simply supplementing germ-free mice with SCFAs in drinking water led to changes in gene expression and histone PTMs that mimicked bacterial colonization. This data strongly suggests that SCFAs are a critical link between the microbiome and epigenetic state. Our understanding of this cross-talk is still in its infancy; thus, discovering how SCFAs interact with host cells on a mechanistic level is crucial to understanding how these systems interact as a whole. Here, we integrate proteomics and metabolomics with biochemical assays to investigate how each SCFA affects epigenetics and cell metabolism. Our results suggest that at physiologic levels, SCFAs induce histone acetylation by activating HATs instead of inhibiting HDACs.

## Results

### Propionate and butyrate induce hyperacetylation in cell culture

To determine the broad effects of extracellular SCFAs on histone PTMs, we treated HCT116 cells with acetate, propionate, and butyrate individually and then performed histone proteomics (*Karch et al., 2013*; *Krautkramer et al., 2015*). This mass spectrometry-based method allows for simultaneous analysis of >70 histone PTMs, which significantly improves on traditional Western blotting (*Thomas et al., 2020*). While the impact of SCFAs on a few of these PTMs has been studied previously, to our

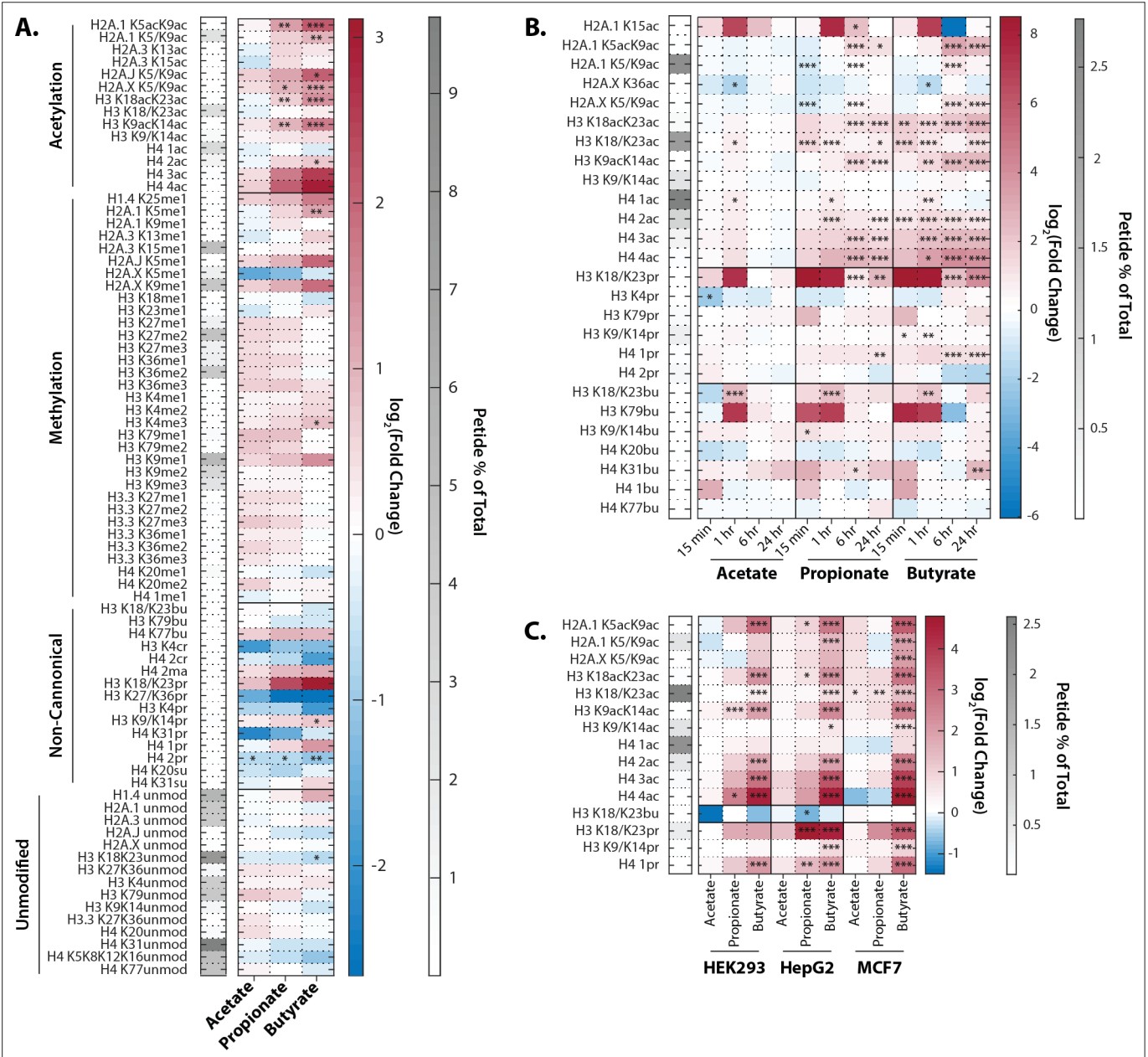

**Figure 1.** Extracellular propionate and butyrate induce histone hyperacetylation. (**A**) Histone proteomics of HCT116 cells treated with 1 mM acetate, propionate, or butyrate for 1 hr. (**B**) Histone acylation over a time course of 1 mM acetate, propionate, or butyrate treatment. (**C**) SCFA treatment (1 mM, 1 hr). of HEK293, MCF7, and HepG2 cell lines. All values are $\log_2$(fold change) over untreated, time-matched controls. *=p≤0.05, **=p≤0.01, ***=p≤0.001. n=3 per condition.

The online version of this article includes the following figure supplement(s) for figure 1:

**Figure supplement 1.** Physiological effects of SCFA treatment.

knowledge this is the first study to show how SCFAs regulate a range of both canonical and non-canonical PTMs (*Hinnebusch et al., 2002*; *Kiefer et al., 2006*; *Silva et al., 2018*; *Wang et al., 2018*). HCT116 cells were initially chosen because they are derived from colon, and colon cells are in most frequent contact with bacterially produced SCFAs.

Propionate and butyrate induced rapid and dose-dependent increases in histone acetylation but did not affect histone methylation (*Figure 1A*). In contrast, acetate had no significant effect on histone PTMs at any dose or time point tested. These results are consistent with previous reports (*Hinnebusch et al., 2002*; *Kiefer et al., 2006*; *Silva et al., 2018*). The hyperacetylation phenotype with propionate and butyrate was rapid and stable, persisting for at least 24 hr (*Figure 1B*). We also detected changes in both histone propionylation and butyrylation on multiple sites, including significant increases in K18/K23 propionylation after both propionate and butyrate treatment. In general, changes in histone propionylation and butyrylation were less consistent across experiments compared to the robust changes in histone acetylation, likely due in part to the technical challenges of measuring low-abundance propionylation/butyrylation. Based on these experiments, we chose a standard treatment of 1 mM SCFA for 1 hr, which we used for further experiments unless otherwise indicated. This dose did not affect media pH, media glucose, or cell viability for at least 24 hr (*Figure 1—figure supplement 1*). The 1 mM dose is within the physiological concentrations in the colon (which range from 0.5 to 15 mM), but is higher than reported concentrations in other parts of the body (which range from 0 to 150 µM) (*Cummings et al., 1987*; *Parada Venegas et al., 2019*).

To determine whether this effect was unique to HCT116 cells, we also treated three other cell lines derived from diverse human tissues. HEK293, HepG2, and MCF7 cells showed remarkably similar hyperacetylation phenotypes after propionate and butyrate treatment (*Figure 1C*). Thus, the ability of propionate and butyrate to increase histone acetylation, but not methylation, can be generalized to multiple cell types.

## Propionate and butyrate are rapidly metabolized into acyl-CoAs

This experiment raised several intriguing questions. First, what differentiates propionate and butyrate from acetate? It is general knowledge that butyrate can act as an HDAC inhibitor, slowing the removal of acetate from chromatin (*Davie, 2003*). This could explain the difference between acetate and butyrate, but it does not necessarily explain the difference between acetate and propionate. To test whether propionate could act as an HDAC inhibitor in HCT116 cells, we performed HDAC assays on nuclear extract and determined the apparent $IC_{50}$ values for these SCFAs and their corresponding acyl-CoA forms. *Table 1* lists the $IC_{50}$ values of HDAC inhibition for propionate, butyrate, and corresponding acyl-CoAs. These $IC_{50}$ values are on the low end of previously published values, which range from 50 to 300 µM in nuclear extract and 1–10 mM in whole cells (*Huber et al., 2011*; *Silva et al., 2018*; *Vinolo et al., 2011*; *Waldecker et al., 2008*). While SCFA concentrations in media were theoretically higher than our determined $IC_{50}$ values (*Figure 2C*), it has been previously reported that intracellular concentrations of SCFAs are much lower than extracellular concentrations, which could have bearing on this mechanism (*Donohoe et al., 2012*).

To determine intracellular SCFA concentrations before and after the extracellular addition of SCFAs, we treated cells with fully labeled $^{13}$C-acetate, propionate, or butyrate and ran targeted LC-MS/MS metabolomics to track SCFAs through cellular metabolism (*Figure 2A*). After treatments, SCFA concentrations in media declined from 1 mM to 0.5 mM over 24 hr. SCFA levels measured in cellular extractions were similar among the three SCFAs (max ~ 3-fold difference), with all showing a decline by 24 hr (*Figure 2C*). From levels determined in the cellular extractions, we estimated intracellular concentrations of ~40 µM for butyrate and ~100 µM for propionate and acetate, which are consistent with previous reports (*Donohoe et al., 2012*). In additional control experiments, SCFA concentrations in

**Table 1.** $IC_{50}$ of HDAC inhibition for SCFAs and acyl-CoAs.

$IC_{50}$ values of HDAC inhibition for propionate, butyrate, propionyl-CoA, and butyryl-CoA. Values are calculated from dose curves using n=2 technical replicates, raw data can be found in *Source data 1*.

| Molecule | $IC_{50}$ |
|---|---|
| Propionate | 223 ± 64 µM |
| Butyrate | 52 ± 11 µM |
| Propionyl-CoA | 18 ± 7 mM |
| Butyryl-CoA | 1.13 ± 0.01 mM |

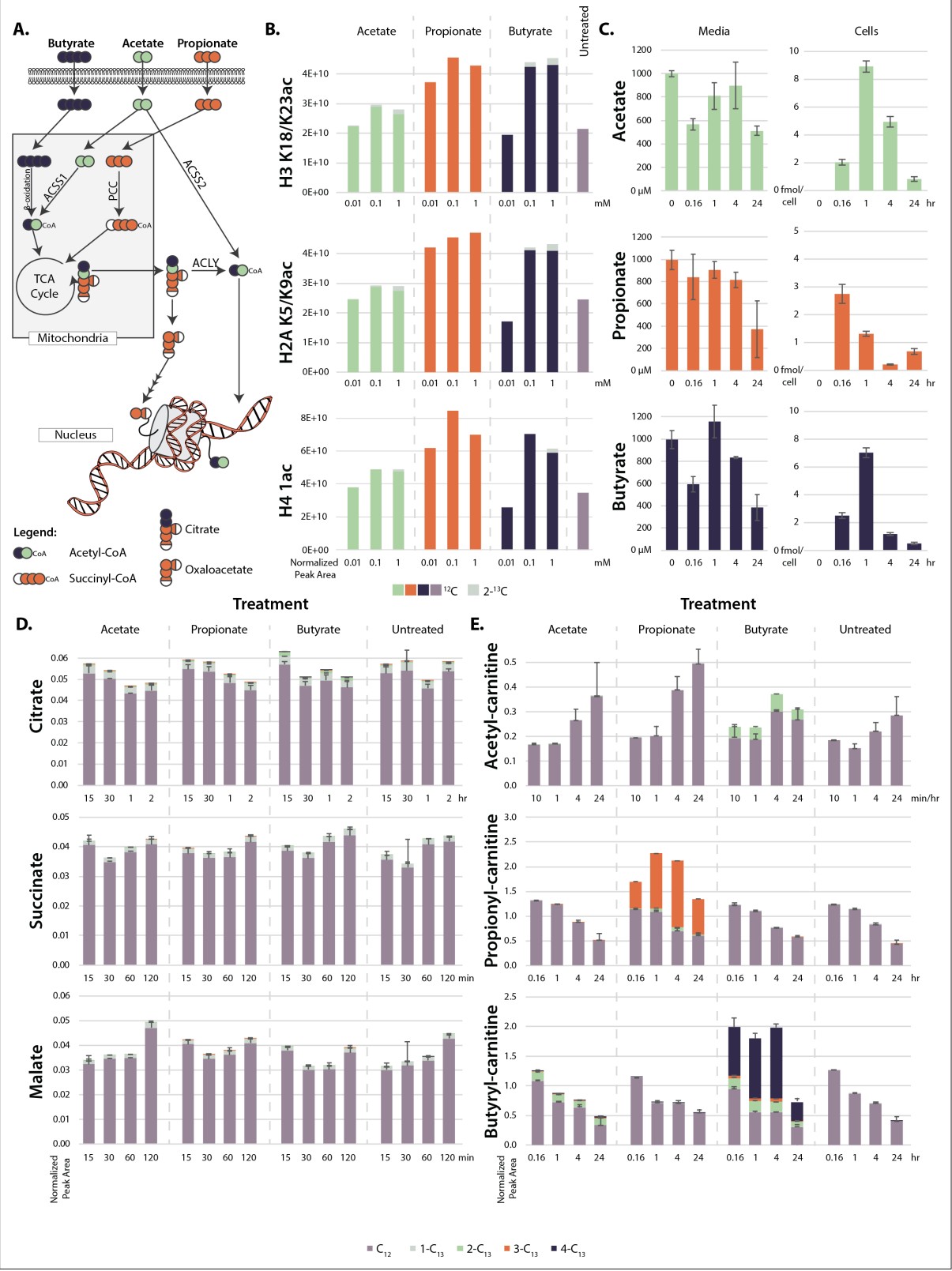

**Figure 2.** Metabolism of extracellular short-chain fatty acids (SCFAs). (**A**) Overall scheme of SCFA metabolism. (**B**) Proportion of [13]C on histones after 4 hr treatment with labeled SCFAs. (**C**) Concentration of SCFAs in media (µM) and in cells (femtomole/cell) over a 24 hr time course. (**D, E**) [13]C labeling of TCA cycle metabolites (**D**) and acyl-carnitines (**E**) over a 24-hr time course. Values are average and standard deviation of normalized signal intensity of ≥3 biological replicates.

*Figure 2 continued on next page*

*Figure 2 continued*

The online version of this article includes the following figure supplement(s) for figure 2:

**Figure supplement 1.** Further SCFA metabolism.

**Figure supplement 2.** Metabolism of SCFAs to acetyl-CoA is not necessary to induce hyperacetylation.

media without cells incubated at 37°C declined at almost an identical rate as that observed in media incubated with cells, suggesting that the bulk of the decrease in media SCFA concentration was due to volatility and not to cellular consumption (*Figure 2—figure supplement 1*). The calculated values suggest that free SCFAs are rapidly metabolized instead of building up in cells (*Boets et al., 2017*; *Clausen and Mortensen, 1994*; *Clausen and Mortensen, 1995*). With this in mind, we searched for labeled carbons in downstream TCA cycle metabolites. However, we found no significant labeling of TCA cycle metabolites after propionate and butyrate treatment (*Figure 2D*). Instead, labeled carbons built up as acyl-carnitine species, appearing in as little as 10 min and remaining stable for 24 hrs (*Figure 2E*). Intriguingly, this buildup did not occur in acetate-treated cells. The same trend occurred in acyl-CoA levels, although these species degrade rapidly during sample processing and are thus difficult to detect by mass spectrometry (*Figure 2—figure supplement 1*). Accumulation of acyl-CoAs after SCFA treatment has been previously reported (*Basu et al., 2011*). In that study, increases in acyl-CoA lead to a decrease of free coenzyme A. Under our conditions, however, free carnitine and free CoA levels did not significantly change (*Figure 2—figure supplement 1*). To confirm that these metabolic changes were consistent across cell lines, we also performed targeted LC-MS/MS metabolomics in HEK293, HepG2, and MCF7 cells. All cell lines showed the same upregulation of acyl-carnitines after butyrate and propionate treatment (*Figure 2—figure supplement 1*). Ultimately, these data suggest that histone hyperacetylation may not be due to the presence of intracellular propionate or butyrate, as the intracellular levels of these metabolites are low, but to the presence of propionyl- or butyryl-CoA. Since propionyl- and butyryl-CoA do not inhibit HDAC at physiological concentrations, this raises the possibility that the histone hyperacetylation phenotype may not be due to HDAC inhibition, but to a different mechanism (*Table 1*; *Vogelauer et al., 2012*).

The intracellular levels of SCFAs do not support robust HDAC inhibition at micromolar treatment concentrations. However, we and others have measured histone hyperacetylation in response to treatment concentrations as low as 10 µM (*Figure 1—figure supplement 1*; *Biermann et al., 2011*; *Donohoe et al., 2012*; *Kespohl et al., 2017*). At these concentrations, HDACs would be inhibited <1%. Alternatively, it has been proposed that hyperacetylation under SCFA treatment is due to a direct mechanism, in which acetyl-CoA generated by SCFAs is added directly onto histones (*Donohoe et al., 2012*). To test whether this occurred in our system, we performed histone proteomics after a 4-hr treatment with $^{13}$C-labeled SCFAs. This treatment only resulted in 1–5% $^{13}$C-labeling on histones, which cannot account for the two- or threefold increases in histone acetylation seen after SCFA treatment (*Figure 2B*). This indicates that the majority of new acetyl groups (~90–95%) were not generated from labeled SCFAs. To confirm that SCFAs do not need to be metabolized into acetyl-CoA to induce hyperacetylation, we knocked down several enzymes involved in acetyl-CoA generation from SCFAs (*Figure 2A*). siRNA knockdown or genetic knockout of ACLY (an enzyme involved in transport of acetyl-CoA out of the mitochondria), ACSS2 (the cytoplasmic acetyl-CoA ligase), and PCC (an enzyme in the pathway that metabolizes propionyl-CoA to succinyl-CoA) had no significant effect on propionate or butyrate-induced histone acetylation (*Figure 2—figure supplement 2*). This data indicates that acyl-CoAs are not metabolized to acetyl-CoA in sufficient quantities to explain the induced hyperacetylation, indicating a different mechanism is at play.

## Propionyl- and butyryl-CoA activate histone acetylation

Next, we tested whether the rapidly formed propionyl- and butyryl-CoA from propionate and butyrate treatment could directly activate HATs (*Table 1*). HAT assays demonstrated that propionyl- and butyryl-CoA induced dose-dependent increases in the HAT activity of nuclear extract (*Figure 3A*). Free propionate and butyrate had no effect. However, since this assay simply measures the release of CoA, the apparent increase in HAT activity could have resulted from increased propionyl- or butyryl-transfer reactions. In addition, these results do not indicate which HAT enzymes may be activated.

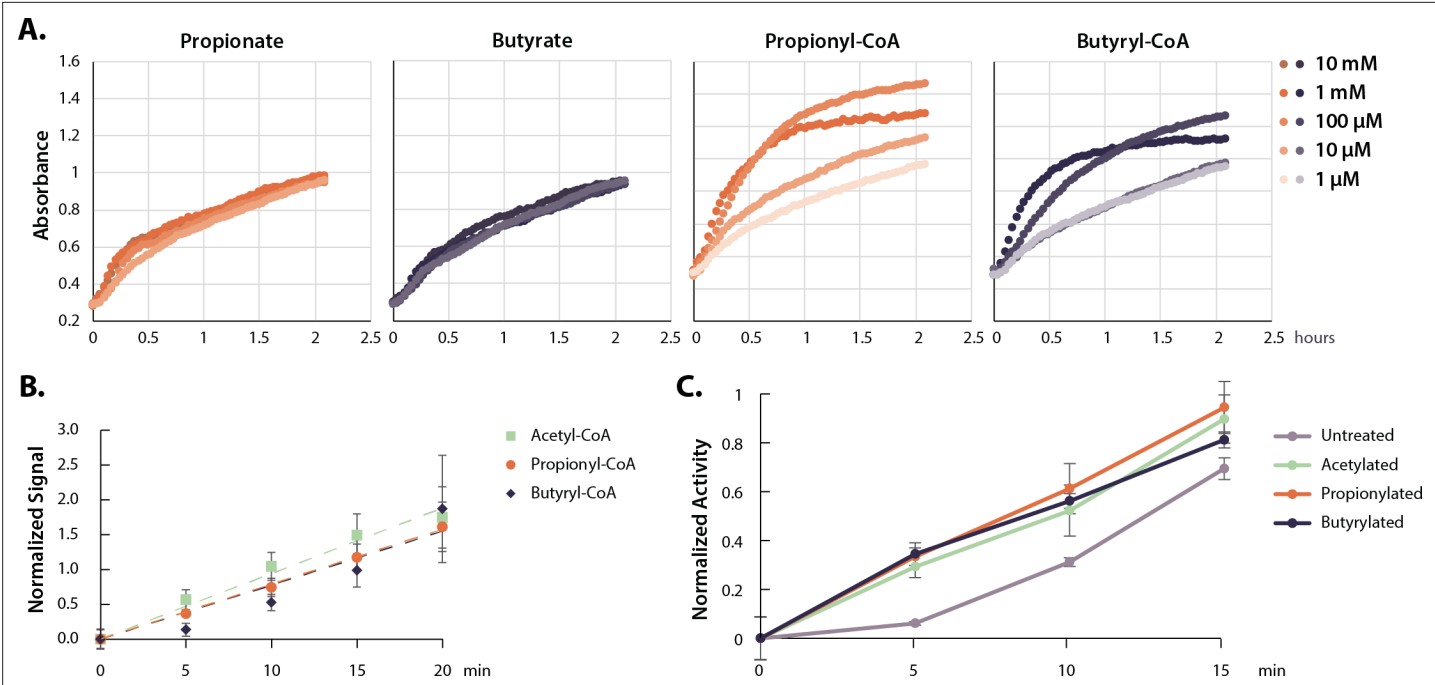

**Figure 3.** Acyl-CoAs activate p300. (**A**) HAT activity of nuclear extract treated with a dose curve of SCFAs and acyl-CoAs. (**B**) Rate of p300 auto-acylation with acetyl-CoA, propionyl-CoA, and butyryl-CoA. Values are average and standard deviation, n=4 per condition. (**C**) Rate of histone acetylation by acetylated, propionylated, or butyrylated p300 using radioactive acetyl-CoA. Values are average and standard deviation, n=3 per condition. Replicate results from this assay performed with different enzyme preparations on different days are shown in *Figure 3—figure supplement 1*. Quantification for all replicates including Western blots and radioactive assays available in *Source data 1*. HAT, histone acetyltransferase; SCFA, short-chain fatty acid.

The online version of this article includes the following figure supplement(s) for figure 3:

**Figure supplement 1.** Recombinant p300 is actively auto-acylated.

HATs can be categorized into three main families—GNAT, MYST, and p300/CBP (*Albaugh et al., 2011*; *Berndsen and Denu, 2008*; *Fan et al., 2015*). Most HATs prefer specific histone lysines and catalyze acetyl-transfer to a handful of histone substrates (*Roth et al., 2001*). However, p300/CBP acetylates a large number of lysine sites in vitro, and is known to target several of our measured hyper-acetylation sites in vivo (*Henry et al., 2013*). p300/CBP are two closely related transcriptional co-activators—large proteins consisting of several domains and even more binding partners and acetylation targets (*Chan and La Thangue, 2001*; *Dancy and Cole, 2015*; *Goodman and Smolik, 2000*). Both proteins encode an autoinhibitory loop (AIL, also called the autoregulatory or activation loop), which must be auto-acetylated to stimulate HAT activity (*Karanam et al., 2006*; *Thompson et al., 2004*). Large-scale mass-spectrometry surveys have identified sites of propionylation and butyrylation on p300, including on several AIL lysines; however, how these modifications are added and whether they have a functional consequence on p300 activity are not known (*Chen et al., 2007*; *Cheng et al., 2009*; *Kaczmarska et al., 2017*). First, we determined whether unmodified p300 can catalyze auto-acylation using propionyl- or butyryl-CoA. Indeed, we found that recombinant p300 performed auto-acylation with propionyl- or butyryl-CoA at rates indistinguishable from auto-acetylation with acetyl-CoA (*Figure 3B*). Auto-acylation was catalytic, as heat-denatured p300 did not show increases in acylation under the same conditions (*Figure 3—figure supplement 1A*). Proteomics analysis confirmed the location of propionylation and butyrylation on several AIL lysine sites under these conditions (*Table 2*).

Next, we investigated if auto-acylation activates p300 toward histone acetylation. To do so, we compared pre-acylated and unmodified forms of recombinant p300 in radioactive HAT assays to measure acetylation rates on a K18/K23-containing peptide substrate. All forms of acylated p300 (acetylated, propionylated, and butyrylated) were significantly more active than an unmodified control (*Figure 3C*). In addition, there were no significant differences in the activity of auto-acetylated, propionylated, or butyrylated p300, which all showed ~ 500% increase in their initial rates compared to unmodified p300. We confirmed these results by performing Western blot HAT activity assays with

**Table 2.** Sites of p300 acylation.

Sites of lysine acetylation, propionylation, and butyrylation on recombinant p300 treated with acyl-CoAs. AIL: autoinhibitory loop, A: acetylation, P: propionylation, B: butyrylation.

| Lysine site (AIL lysines in bold). | Acetylation | Propionylation | Butyrylation |
|---|---|---|---|
| K1542 | A | P | |
| K1546 | A | P | |
| K1549 | A | P | |
| K1550 | A | | |
| K1551 | A | P | |
| K1554 | A | P | B |
| K1555 | A | P | B |
| **K1558** | A | P | B |
| **K1560** | A | P | B |
| **K1568** | A | P | |
| **K1569** | A | P | |
| **K1570** | A | P | |
| K1583 | A | | |
| K1590 | A | | |
| K1637 | A | P | |

full-length histone H3/H4 dimers/tetramers, which are a more physiological substrate than a single peptide (*Figure 3—figure supplement 1C-D*). Again, these data showed an increase in activity for all three forms of acylated p300. In all, these data demonstrate that p300 catalyzes auto-butyrylation and propionylation on AIL sites, and that auto-acylation activates the enzyme in a similar manner to auto-acetylation.

To investigate whether this previously unappreciated mechanism functions in vivo, we used pharmacological inhibitors of each potential pathway. To minimize any potential effects from HDAC inhibition in these experiments, we used treatment concentrations from 100 to 250 μM, which should lead to no more than 5–15% HDAC inhibition (*Table 1*). First, we treated cells with a potent inhibitor of p300, A485. A485 inhibits both histone acetylation and auto-acetylation of p300, and thus should inhibit histone hyperacetylation due to p300 activation (*Lasko et al., 2017*). Indeed, A485 treatment reversed SCFA-induced hyperacetylation on p300 targets H3K27 and H3K18 (*Figure 4A*, *Figure 4—figure supplement 1A*). A485 treatment did not just reverse SCFA-induced hyperacetylation on histone substrates of p300, but also on the non-histone p300 substrate p53 (K382). Second, we treated cells with the potent HDAC inhibitor, SAHA. Both SAHA and butyrate are thought to be competitive inhibitors of HDACs, although the IC$_{50}$ of SAHA is significantly lower (~10 nm) (*Marks and Breslow, 2007*; *Sekhavat et al., 2007*). Treatment with 10 μM of this potent pan-HDAC inhibitor should effectively saturate HDAC inhibition, at least of class I and II HDACs, which are known to target these sites (*Caslini et al., 2019*; *Kelly et al., 2018*; *Rajan et al., 2018*). Thus, if propionate and butyrate act solely as HDAC inhibitors, they should not induce further hyperacetylation under these conditions. However, 100 μM treatment of propionate or butyrate with SAHA still led to significant increases in p53K382 acetylation over a SAHA control (*Figure 4A*, *Figure 4—figure supplement 1A*). Collectively, these results suggest that propionate and butyrate induce histone hyperacetylation by an alternative mechanism to HDAC inhibition, likely through the auto-activation of p300 by rapidly formed acyl-CoAs.

To corroborate this novel mechanism of SCFA action on chromatin, we transfected HCT116 cells with two full-length p300 constructs: either wild-type (WT) p300 or an AIL mutant in which 11 of the 17 AIL lysines were mutated to a glutamate (ΔGlu) (*Ortega et al., 2018*). The ΔGlu mutation changes

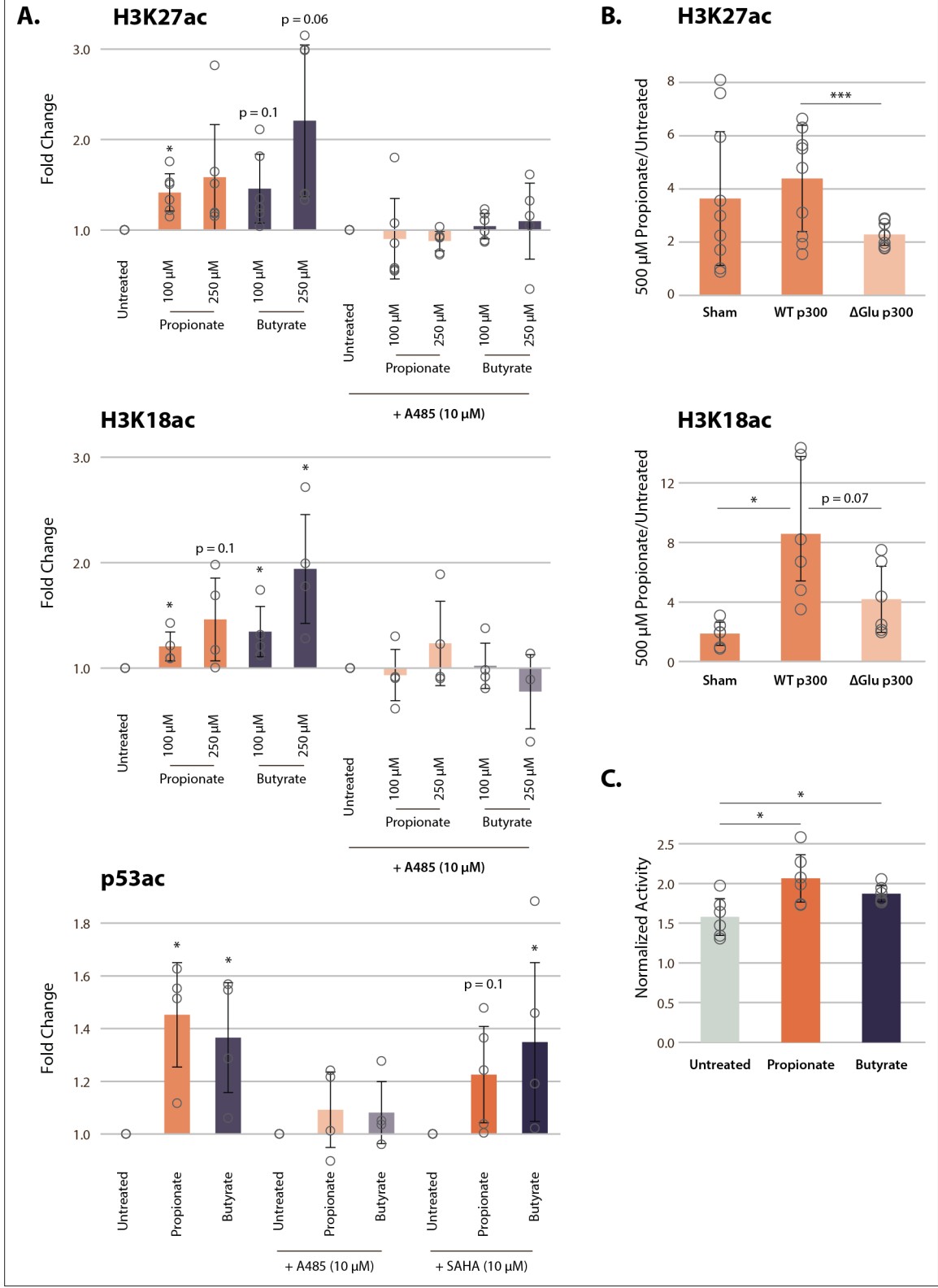

**Figure 4.** p300 inhibition, but not HDAC inhibition, reverses SCFA-induced hyperacetylation. (**A**) Acetylation of H3K27ac, H3K18ac, and p53K382ac after treatment with A485, SAHA, and 100–250 μM of propionate/butyrate for 24 hr. Values are normalized to total H3 or total p53 before calculating fold changes to the appropriate untreated control. (**B**) Acetylation of H3K27ac and H3K18ac in cells transfected with sham, WT p300, or ΔGlu p300 plasmids. Values are fold change over untreated cells with the same transfection. (**C**) Activity of immunoprecipitated p300 after treatment with 500 μM

*Figure 4 continued on next page*

*Figure 4 continued*

of propionate or butyrate. Activity is measured with a radioactive assay and normalized to concentration of immunoprecipitated p300 in each sample. *=p≤0.05, **=p≤0.01, ***=p≤0.001. n≥3 per condition. HDAC, histone deacetylase; SCFA, short-chain fatty acid.

The online version of this article includes the following figure supplement(s) for figure 4:

**Figure supplement 1.** Representative Western blots corresponding to *Figure 4*.

the charge on AIL sites, which in turn blocks auto-acylation of the AIL and prevents p300 activation. As expected, 500 µM propionate treatment induced significant increases in histone acetylation after sham and WT transfections (*Figure 4B*, *Figure 4—figure supplement 1B-C*). However, cells transfected with the ΔGlu mutant showed significantly smaller increases in acetylation, approximately half of that seen in WT-transfected cells. In addition, we suspect that even the small increases in acetylation in the ΔGlu condition were due to residual amounts of endogenous p300 in transfected cells, or possibly acylation of residual lysine sites on the AIL. Nevertheless, these data strongly suggest that the AIL lysines of p300 are necessary to induce the full extent of histone hyperacetylation after propionate treatment.

Finally, we performed immunoprecipitation of full-length endogenous p300 from HCT116 cells treated with 500 µM propionate or butyrate. p300 immunoprecipitated from propionate and butyrate-treated cells showed small but significant increases in activity in a radioactive HAT assay (*Figure 4C*). Corresponding western blots also identified increased propionylation and butyrylation of p300 under these conditions (*Figure 4—figure supplement 1D*). In all, our results demonstrate that p300 activation, rather than HDAC inhibition, plays a critical role for propionate and butyrate-induced chromatin hyperacetylation at low concentrations.

## Discussion

For decades, butyrate and propionate's ability to alter chromatin have been attributed to HDAC inhibition (*Candido et al., 1978*). However, the $IC_{50}$ values for these two molecules are quite high, with reported values ranging between 50 and 300 µM in nuclear extract and between 1 and 10 mM in whole cells (*Huber et al., 2011*; *Silva et al., 2018*; *Vinolo et al., 2011*; *Waldecker et al., 2008*). While it is possible to reach those concentrations in the gut lumen or portal vein, few other organs would normally reach these values. In addition, our previous data suggests that SCFAs can induce histone hyperacetylation in colon, liver, and adipose tissue, which experience SCFA concentrations that differ by orders of magnitude (*Krautkramer et al., 2016*). Even in the gut, intracellular concentrations of SCFAs could be significantly lower than exterior concentrations (*Donohoe et al., 2012*; *Sengupta et al., 2006*). Thus, the idea that increased histone acetylation is primarily mediated by HDAC inhibition requires re-evaluation.

In this study, we report a previously unappreciated mechanism and provide compelling evidence that butyrate and propionate activate p300 at low levels, through the rapid conversion to propionyl- and butyryl-CoA, catalytic auto-acylation, and subsequent activation of p300. While the $IC_{50}$ values for HDAC inhibition are in the high µM to mM range, $K_m$ values for p300 auto-acetylation lie in the nM to low µM range (*Karanam et al., 2006*; *Liu et al., 2008*). Hyperacetylation induced by high doses of butyrate (2–5 mM) are almost immediately reversed after butyrate withdrawal (*Prasad and Sinha, 1976*; *Wang et al., 2018*). In contrast, auto-acetylation of p300 increases its half-life from 4.5 to >24 hr (*Jain et al., 2012*). Thus, even after extracellular SCFAs have been fully metabolized, modified p300 could still theoretically induce hyperacetylation. Auto-acylation of p300 may thus prove a primary mechanism in physiological systems where SCFA concentrations are low.

It is important to note that acetate had negligible effect on histone acetylation in the cell-based systems described in this study. Acetate treatment has been previously reported to increase histone acetylation, but only under extreme conditions such as hypoxia (*Bulusu et al., 2017*; *Gao et al., 2016*), or after genetic manipulation (*Zhao et al., 2016*). Others have reported no effect of acetate in conditions similar to those described here (*Hinnebusch et al., 2002*). This phenomenon has traditionally been explained by the fact that acetate is a worse HDAC inhibitor than other SCFAs (*Hinnebusch et al., 2002*; *Waldecker et al., 2008*). Based on the results reported here, acetate's lack of effect can be explained by the fact that acetate is not rapidly metabolized to acetyl-CoA, as propionate and butyrate are to their respective acyl-CoAs. It has been previously shown that extracellular acetate

concentration does not necessarily correlate with intracellular concentrations of acetyl-CoA. Instead, under normal conditions, acetyl-CoA is mainly generated in the mitochondria from glucose (*Wellen et al., 2009*). Since there is no evidence to date of acetyl-CoA crossing the mitochondrial membrane, acetyl-CoA must be shuttled to the cytosol via citrate. Once in the cytosol, citrate is converted into oxaloacetate and acetyl-CoA by ATP citrate lyase (ACLY). However, stress conditions such as hypoxia, or genetic knockout of ACLY, can cause the cell to rely on acetate for acetyl-CoA production (*Bulusu et al., 2017*; *Gao et al., 2016*; *Zhao et al., 2016*). We do not expect these conditions to exist in the experiments described here.

The main pathways responsible for butyryl- and propionyl-CoA synthesis in the nucleus and cytoplasm are still unclear (*Corfe, 2012*; *Fujino et al., 2001*; *Luong et al., 2000*; *Trefely et al., 2020*). However, our results suggest that p300 auto-acylation and activation via SCFAs depend on robust cellular synthesis of the corresponding acyl-CoAs. In vivo, synthesis of acyl-CoAs may depend on complex interactions between various SCFAs. While we investigated each SCFA individually, it is possible that mixtures of each SCFA could have varied effects on histone acetylation. However, there is some data to suggest that in cells treated with mixtures of SCFAs, acetylation increased based only on the sum of butyrate and propionate concentrations (*Kiefer et al., 2006*). This is consistent with our observation that the use of normal or dialyzed fetal bovine serum (FBS) in cell culture media did not affect the histone hyperacetylation phenotype (dialyzed FBS does not contain acetate). We predict that rapid histone acetylation, as described for the mechanism revealed in this study, will depend on the ability of cells to convert each SCFA to its corresponding acyl-CoA.

The p300 activation mechanism revealed here could also resolve outstanding issues. SCFA treatment phenocopies other HDAC inhibitors in some cases (*Chang et al., 2014*; *Rahman et al., 2003*; *Zhou et al., 2011*), but not others (*Milton et al., 2012*; *Siavoshian et al., 2000*; *Zhao et al., 2020b*). An alternate method for SCFAs to regulate histone acetylation could account for some of these discrepancies. In addition, several groups have reported that certain phosphatase or PKC inhibitors can reverse SCFA-induced hyperacetylation (*Cuisset et al., 1997*; *Cuisset et al., 1998*; *Rickard et al., 1999*). While it has been proposed that this is due to phosphorylation of HDACs (*Davie, 2003*), it is interesting to note that these inhibitors are also known to interact with p300 (H.-H. *Cheng et al., 2014*; *Granja et al., 2008*; *Zgheib et al., 2012*).

Our proposed mechanism also raises the question: why is p300 specifically activated after SCFA treatment? Many HATs can acylate histones, and some are even known to auto-acetylate (*Kaczmarska et al., 2017*; *Leemhuis et al., 2008*; *Yang et al., 2012*). In general, rates of acylation decrease as acyl chains increase, to the extent that butyryl-CoA can act as a competitive inhibitor for some HATs (*Carrer et al., 2017*; *Montgomery et al., 2015*; *Ringel and Wolberger, 2016*; *Simithy et al., 2017*). Thus, a sharp increase in acyl-CoA concentrations could slow most HATs, which cannot efficiently use longer acyl-CoAs as substrates. Intriguingly, previous reports indicate that p300 is also less efficient at histone propionylation and butyrylation than it is at histone acetylation (*Kaczmarska et al., 2017*), which is consistent with our histone PTM analyses (*Figure 1*). However, our measured rates of p300 auto-acylation were indistinguishable between acetate, propionate, and butyrate. In addition, the reported $K_m$ of p300 for auto-acetylation (~100 nM) is lower than for acetylation of histone substrates (~6 µM) (*Karanam et al., 2006*; *Liu et al., 2008*). While the differences between the catalytic mechanisms for p300 auto-acylation and histone acylation have not yet been elucidated, this data suggests that p300 may have different substrate preferences for these two reactions. Thus, p300 may prefer to use acyl-CoAs to auto-acylate, which would activate the enzyme toward other substrates targeted for acetylation. In that case, while higher concentrations of acyl-CoAs may slow or inhibit other HATs, they could preferentially activate p300 via auto-acylation. The fact that our proteomics data shows more robust increases in histone acetylation than histone propionylation or butyrylation would further support this hypothesis (*Figure 1*).

It should be noted that SCFAs are also known to bind and activate several G-protein coupled receptors (GPCRs), especially FFAR2, FFAR3, and HCAR2 (*Brown et al., 2003*; *Inoue et al., 2014*; *Priyadarshini et al., 2018*; *Singh et al., 2014*; *Thangaraju et al., 2009*; *Xiong et al., 2004*). These receptors are not expressed in most of the cell lines investigated here, and do not influence the hyperacetylation phenotype when they are (*Figure 1C*). However, exploring the potential interactions between GPCR activation and histone hyperacetylation will be critical to a full understanding of SCFA signaling and metabolism, and its impact on human health.

Finally, we are not suggesting that SCFAs cannot induce HDAC inhibition under certain physiological conditions. Butyrate especially is likely capable of inducing HDAC inhibition in the colon, where extracellular concentrations of butyrate are often >10 mM. In fact, this may be the reason that butyrate leads to higher levels of histone acetylation than propionate at the 1 mM treatment concentration (*Figure 1*), but not at the 100 µM treatment concentration (*Figure 4*). Discovering which mechanism is most prevalent at various concentrations may assist in understanding the pharmacology of this crucial molecule.

The results presented in this study provide key new insight into the actions of normal physiological and therapeutic levels of SCFAs. This is especially important in the case of butyrate, which is the subject of over 100 active clinical trials. In fact, there is already evidence that low SCFA concentrations can have widely different physiological effects while still increasing histone acetylation (*Biermann et al., 2011*; *Donohoe et al., 2012*). By comparing cellular responses to high and low SCFA doses (e.g., 5 mM and 50 µM), it may be possible to explore distinct cellular responses given the different mechanisms. It may also be necessary to reexamine the systemic effects of SCFAs. Many studies use high doses of SCFAs (5–30 mM), even in regions of the body where physiological concentrations should be thousands of times lower. It is possible that these doses are not necessary to induce hyperacetylation and could in fact obscure biologically relevant pathways. In the end, this new data suggest that the mechanism of action extends beyond HDAC inhibition and should be incorporated into the rationale for the therapeutic use of SCFAs.

# Materials and methods

## Key resources table

| Reagent type (species) or resource | Designation | Source or reference | Identifiers | Additional information |
|---|---|---|---|---|
| HCT116 cell line (*Homo-sapiens*) | Colorectal carcinoma | ATCC | CCL-247 | RRID:CVCL_0291 |
| HEK-293 cell line (*Homo-sapiens*) | Kidney epithelial | ATCC | CRL-1573 | RRID:CVCL_0045 |
| HepG2 cell line (*Homo-sapiens*) | Liver epithelial | ATCC | HB-8065 | RRID:CVCL_0027 |
| MCF7 cell line (*Homo-sapiens*) | Adenocarcinoma epithelial | ATCC | HTB-22 | RRID:CVCL_0031 |
| PCCA mutant cell lines (*Homo-sapiens*) | Propionic acidemia cell lines | NIGMS Human Genetic Cell Repository, Coriell Institute | GM00371 GM00405 GM22010 GM22011 GM22123 GM22126 | |
| Transfected construct (human) | siRNA to ACSS2 (SMARTpool) | Dharmacon | L-010396-00-0005 | |
| Transfected construct (human) | siRNA to ACLY (SMARTpool) | Dharmacon | L-004915-00-0005 | |
| Transfected construct (human) | siRNA to PCCA (SMARTpool) | Dharmacon | L-008965-00-0005 | |
| Transfected construct (human) | p300 mutant constructs | Ortega et. al. 2018 | WT p300, ΔGlu p300 | |
| antibody | ACLY, rabbit | CST | 4,332 S | RRID:AB_2223744, 1:1,000 dilution |
| antibody | ACSS2, mouse | Novus Biologicals | NBP2-01269 | 1:1,000 dilution |
| antibody | FFAR2/GPR43, mouse | Novus Biologicals | MAB10082-100 | 1:1,000 dilution |
| antibody | FFAR3/GPR41, rabbit | Novus Biologicals | NBP2-14014 | 1:1,000 dilution |
| antibody | H3, mouse | Abcam | ab24834 | RRID:AB_470335, 1:5,000 dilution |
| antibody | H3 K18ac, rabbit | Abcam | ab1191 | RRID:AB_298692, 1:1,000 dilution |
| antibody | H3 K27ac, mouse | Active Motif | 39,685 | RRID:AB_2793305, 1:1,000 dilution |

*Continued on next page*

*Continued*

| | | | | |
|---|---|---|---|---|
| antibody | HA-Tag, rabbit | CST | 3,724 | RRID:AB_1549585, 1:1,000 dilution |
| antibody | p300 (NM11), mouse | SCBT | sc-32244 | RRID:AB_628076, 1:1,000 dilution |
| antibody | p53, mouse | CST | 2,524 | RRID:AB_331743, 1:1,000 dilution |
| antibody | p53 K382ac, rabbit | CST | 2,525 | RRID:AB_330083, 1:1,000 dilution |
| antibody | Pan anti-acetyllysine, rabbit | CST | 9,814 | 1:1,000 dilution |
| antibody | Pan anti-butyryllysine, mouse | PTM Biolabs | PTM-329 | 1:500 dilution |
| antibody | Pan anti-propionyllysine, rabbit | PTM Biolabs | PTM-201 | 1:500 dilution |

## Materials

Cell culture reagents were purchased from Invitrogen (Carlsbad, CA). All reagents used for proteomic and metabolomic samples were HPLC-grade. All cell lines were regularly tested for mycoplasma contamination.

## SCFA cell line treatments

All cell lines were plated in Minimum Essential Media (MEM) supplemented with 10% dialyzed FBS the day before SCFA treatment. MEM contains ~5 mM glucose, compared to ~25 mM in Dulbecco's modified Eagle's medium. Stock solutions of each SCFA were prepared fresh in PBS the day of each experiment. After collection, cell pellets and metabolite extracts were stored at – 80°C. Using preliminary data sets to estimate population variance, a minimum of three biological replicates are necessary to detect changes of 50% or greater, assuming $\alpha=0.05$ and power=0.80. We thus chose 3–4 replicates for all experiments unless otherwise noted.

### Histone proteomics

We performed histone proteomics as described previously (*Krautkramer et al., 2016*; *Sidoli et al., 2016*). The method is briefly described below.

### Histone extraction and derivatization

Cells were trypsinized and washed once with PBS. Cell pellets were then dounce homogenized in lysis buffer (10 mM Tris-HCl, 10 mM NaCl, 3 mM MgCl$_2$, and pH 7.4)+ protease inhibitors (10 mM nicotinamide, 1 mM sodium butyrate, 4 µM trichostatin A, 10 µg/ml leupeptin, 10 µg/ml aprotinin, and 100 µM PMSF) and nuclei were spun out. Histones were extracted from the nuclei pellet with 0.4 N H$_2$SO$_4$ for 4 hr. Extracted histones were then TCA precipitated overnight and washed with acetone. After performing a Bradford assay to determine protein concentrations, 5 µg of each sample was dried down and resuspended in 100 mM triethylammonium bicarbonate buffer. Unmodified lysines were labeled with d$_6$-acetic anhydride, a labeling method that allows for quantification of histone acylations (*Thomas et al., 2020*). Labeled histones were digested with trypsin and labeled with N-termini phenyl isocyanate before stage tip desalting and resuspension sample diluent ( 5% ACN, 0.1% acetic acid in H$_2$O).

### UPLC MS/MS analysis

Derivatized histone peptide was injected onto a Dionex Ultimate3000 nanoflow HPLC with a Waters nanoAcquity UPLC C18 column (100 m × 150 mm, 3 m) coupled to a Thermo Fisher Q-Exactive mass spectrometer at 700 nl/min. Mobile phase consisted of water + 0.1% formic acid (A) and acetonitrile + 0.1% formic acid (B). Histone peptides were resolved with a two-step linear gradient of 5–35% B over 45 min followed by 35–95% B over 10 min. Data was acquired in data-independent acquisition (DIA) mode. The MS1 scan resolution=35,000, automatic gain control target=1 ×10$^6$, and scan range=390–910 *m/z*, followed by a DIA scan with a loop count of 10. DIA settings: window

size=10 $m/z$, resolution=17,500, automatic gain control target=1 ×10$^6$, DIA maximum fill time=Auto, and normalized collision energy=30. For each cycle, 1 full MS1 scan was followed by 10 MS2 scans using an isolation window size of 10 $m/z$.

## Data analysis

Thermo RAW files were loaded into EpiProfile 2.0 and run using the AcD3 module (*Yuan et al., 2018*). Normalization and statistics were then performed using our previously published Histone Analysis Workflow (*Thomas et al., 2020*). RAW files, peak tables, and processing scripts have been uploaded to the MassIVE database and can be accessed with username: MSV000087800_reviewer and password: SCFAp300.

## Metabolomics

### Sample preparation

Cells were plated in six-well plates and allowed to reach ~ 70% confluency. Media was replaced with fresh media 1 hr before the experiment, and then again with media+SCFAs at time 0. Extraction and derivatization solutions were also prepared at least 1 hr before the experiment and stored at – 20°C. Extraction solution contained 80:20 methanol:water with 1:2500 dilution of a heavy amino acid mix standard (MSK-A2, Cambridge Isotope Laboratories); derivatization solution is described below. At each time point, media was collected for SCFA measurements (see below) and cells were washed 3× with 0.9% NaCl. Then, 700 µl of ice-cold extraction solution was added and plates were set at – 80°C for 15 min. After 15 min, cells were scraped into extraction solution and aliquots were taken for SCFA measurement and DNA quantification. The rest was spun down at max speed for 5 min at 4°C. The resulting supernatant was split in two and dried down using a speed vac and stored at – 80°C. Samples were reconstituted in ice-cold 50 mM ammonium acetate and spun down again before injection into the mass spec. One half of the sample was run in positive mode, while the other half was run in negative mode (details below). Samples were reconstituted in batches and run in random order to minimize error due to metabolite decay or instrumentation.

SCFA concentrations were measured by LC-MS using the derivatization protocol described by *Lu et al., 2013*. In brief, 10 µl of media or cell extract was added to 150 µl of derivatization solution containing 0.5 mM of 2-hydrazinoquinoline, triphenylphosphine, 2,2′dipyridyl disulfide, and either 250 µM (for media) or 25 µM (for cells) unlabeled acetate, propionate, and butyrate. Samples were incubated at 60°C for 1 hr and then spun at max speed for 5 min to remove any cellular debris. The resulting supernatant was mixed 1:1 with ice-cold 50 mM ammonium acetate before injection.

To measure cellular concentration, we used the Hoechst staining protocol described by *Muschet et al., 2016*. To do so, 20 µl of metabolite extract was added to 80 µl of Hoechst 33342 (CST) diluted to 20 µg/ml in ddH$_2$O. Samples were incubated in the dark for 30 min in a black 96-well plate and then read on a plate reader with Ex/Em=355/465 nm. Samples were compared to a dose curve of known cell numbers.

### UPLC MS/MS analysis

All metabolites were injected onto a Dionex Ultimate3000 nanoflow HPLC with a Waters ACQUITY UPLC BEH C18 column (2.1 mm×100 mm) coupled to a Thermo Fisher Q-Exactive mass spectrometer at 0.2 ml/min. Separate columns were used for positive and negative mode to reduce TBA contamination. For samples run in positive mode, mobile phase consisted of water +5 mM ammonium acetate + 0.05% acetic acid (A) and 90% acetonitrile +5 mM ammonium acetate + 0.05% acetic acid (B). Positive mode metabolites were resolved with a two-step linear gradient of 2–85% B over 5 min followed by 85–95% B over 6 min. For derivatized samples, data was acquired in data-dependent acquisition (DDA) mode. MS1 scan resolution=140,000, automatic gain control target=3×10$^6$, and scan range=189–450 $m/z$. MS2 scan resolution=35,000, automatic gain control target=1× 10$^5$, loop count=5, and isolation window=4.0 $m/z$. For other positive mode samples, only MS1 data was acquired in two isolation windows from 60 to 184 $m/z$ and 189 to 1200 $m/z$. MS1 scan resolution=70,000, automatic gain control target=1× 10$^6$.

For samples run in negative mode, mobile phase consisted of methanol (A) and water + 3% methanol +10 mM tributylamine + 0.05% acetic acid (B). Metabolites were resolved with a linear gradient

of 95–5% B over 15 min. Only MS1 data was acquired. Scan resolution=70,000, automatic gain control target=$1\times10^6$, and scan range=59–880 $m/z$.

### Data analysis

Metabolite identification and peak integration were performed in EL-MAVEN (Elucidata). For SCFA concentrations, heavy SCFAs were normalized to unlabeled internal controls and SCFA dose curves run throughout the course of the experiment. For all other metabolites, data was normalized to heavy amino acid standards and cell number determined by Hoechst stain. All $^{13}$C-labeling was corrected based on natural $^{13}$C abundance.

## ACLY, ACSS2, and PCC knockdowns

HCT116 cells were transfected with siRNA SMARTpools targeting ACLY, ACSS2, and PCC according to the manufacturer's protocols. In short, 10 nM ACLY, ACSS2, and non-targeting siRNAs and 20 nM PCC siRNA were transfected using Lipofectamine RNAiMAX and opti-MEM media and allowed to incubate for 48 hr. Media was then replaced with fresh MEM media containing 1 mM SCFA and incubated for 1 hr. Knockdown efficiency was assessed by Western blotting.

Primary lymphoblasts from patients with propionic acidemia and family matched controls were purchased from the NIGMS Human Genetic Cell Repository. Propionic acidemia is a rare genetic disease caused by mutations in PCC. Knockout of PCC activity was confirmed using the HPLC assay developed by Y.-N. *Liu et al., 2016*. Since lymphoblasts are grown in suspension, stock solutions of SCFA were added directly to media, resulting in 1 mM final concentrations. Cells were treated with SCFAs for 4 hr before collection.

## HDAC assay

HDAC activity of HCT116 nuclear extract was assayed using the Fluorometric HDAC Activity Assay Kit from Abcam (ab156064) according to the manufacturer's protocol. Briefly, nuclear extract was prepared by dounce homogenizing cell pellets in lysis buffer (10 mM Tris-HCl, 10 mM NaCl, 3 mM MgCl$_2$, and pH 7.4), then spinning out nuclei and resuspending in nuclear extraction buffer (50 mM HEPES, 420 mM NaCl, 0.5 mM EDTA, 0.1 mM EGTA, 10% glycerol, and pH 7.5). Protein concentration was determined by Bradford, and then 50 µg was added to dose curves of butyrate, propionate, and their corresponding acyl-CoAs. TSA was used as a positive control. Fluorescence signal (Ex/Em=360/450 nm) was read on a plate reader at 2 min intervals for 1 hr. Values were averaged over two technical replicates. IC$_{50}$ values were calculated using GraphPad Prism.

## HAT assays

### Colorimetric assay

HAT activity was assayed using the HAT Activity Assay Kit from Sigma-Aldrich (EPI001), which measures release of free CoA over time. Acetyl-CoA levels were kept constant at 100 µM. Nuclear extract was the same as that used in the HDAC assay. 50 µg nuclear extract was added to kit components and dose curves of butyrate, propionate, and corresponding acyl-CoAs. Signal at 440 nm was read on a plate reader at 2 min2-min intervals for 2 hr. Values are averaged over two technical replicates.

### *Recombinant p300—*

Recombinant catalytic domain of p300 was purchased from Enzo Life Sciences (BML-SE451). To remove endogenous acetylation from the recombinant protein, p300 was incubated with mammalian SIRT2 and NAD. SIRT2 was kindly provided by Dr. Mark Klein. For each experiment, equimolar amounts of SIRT2 and p300 and a tenfold molar excess of NAD were incubated at RT for 45 min. This incubation reduced endogenous acetylation on p300 five- to tenfold without inhibiting auto-acylation (*Figure 3—figure supplement 1*). Excess NAD was then removed by running protein over a Zeba Spin desalting column (Thermo Fisher Scientific). Acyl-CoAs were then added back to the enzyme and auto-acylation was allowed to continue at RT for 30–60 min. For untreated controls, the same amount of CoA was added to account for CoA concentration. Finally, for denatured samples, p300 was incubated with SIRT2 and then boiled at 95°C for 5 min before the addition of acyl-CoAs.

## Auto-acylation assay

Recombinant enzyme was treated with SIRT2, then diluted to 1 µM in HAT assay buffer (50 mM Bis-Tris, 25 mM Tris, 10 mM NaCl, and pH 7) with 80 µM acyl-CoA. For each time point, 2 µl of each sample was dotted directly onto nitrocellulose membrane. Membranes were then allowed to dry for 30 min before blocking and incubating with antibodies specific to each acyl-CoA.

## Identification of lysine acylation

Recombinant enzyme was prepared as described above, then dried down, resuspended in 100 mM triethylammonium bicarbonate buffer, and trypsinized overnight. Samples were then stage tip desalted and resuspended in sample diluent (same as histone protocol above). Samples were run on a Thermo Fisher Q-Exactive mass spectrometer using the same settings as histone samples, except that they were run in DDA rather than DIA mode. Acyl identification was performed using Mascot (Matrix Science) and manually validated in XCalibur (Thermo Fisher Scientific). Acylation had to be present in two of three samples to be reported.

## $^{3}$H acetylation assay

Tritiated acetyl-CoA was purchased from PerkinElmer. For each assay, reaction buffer was made by using HAT assay buffer with 5 µM cold acetyl-CoA, 1 µM hot acetyl-CoA, and 25 µM biotinylated peptide. Biotinylated peptide was kindly provided by Dr. Vyacheslav Kuznetsov and corresponds to resides 9–23 of histone H3. Reactions were started by addition of 24 µl reaction buffer to 2 µl acetylated p300 (final concentration 10 nM). At each time point, 7 µl of the reaction mixture was added to 150 µl of cold 2% TFA + 5% Streptavidin Sepharose High Performance (v/v, Cytiva). After the time course was completed, samples were incubated at RT with shaking for 30 min. The samples were then spun down at 1000×$g$ for 1 min, and then washed with 2 ml of 10% acetone. The washed sepharose was then directly added to 2 ml Ultima Gold scintillation cocktail (PerkinElmer). Radioactivity was read by a Tri-Carb 2910TR scintillation counter (PerkinElmer). The scintillation counter was calibrated before each run.

## Western HAT assays

For western samples, 100 nM acylated p300 was added to a master mix of 100 µl acetyl-CoA and 0.4 µg/µl histone H3/H4 dimers. Samples were collected by adding 20 µl directly to 5 µl of 4× sample loading buffer (LI-COR) and heating at 95°C for 5 min. Samples were run on Bolt 4–12% Bis-Tris gels using the XCell II blot module (Thermo Fisher Scientific). Protein was transferred onto nitrocellulose membranes for 45 min and then allowed to block in 5% BSA for 1 hr. Samples were incubated in primary antibody overnight, before washing and incubating in LI-COR secondary antibody for 1 hr. All membranes were imaged on a LI-COR Odyssey imager and quantified using ImageStudio (LI-COR). Whenever possible, antibodies for modifications and loading controls were run simultaneously using two separate fluorescent LI-COR channels. *p300 transfections*—p300 plasmids were a kind gift from Dr. Daniel Panne (*Ortega et al., 2018*) HCT116 cells were grown to ~ 60% confluency and transfected with Lipofectamine 3000 according to the manufacturer's protocols. After 48 hr, media was replaced with MEM media containing 500 µg/ml Geneticin (Thermo Fisher Scientific). Antibiotic media was then replaced every 2 days for 1 week to select for transfected cells. After 1 week, cells were treated with 500 µM propionate for 2 hr. Samples were collected as described previously. HA and p300 blots were performed as described in the previous section, except that transfers were allowed to run overnight to maximize signal for high molecular weight proteins. *p300 immunoprecipitation*—HCT116 cells were grown to 60–70% confluency and treated with 500 µM propionate or butyrate for 1 hr. Cells were lysed using RIPA buffer and Pierce protease phosphatase inhibitors (Thermo Fisher Scientific). Protein concentrations were determined by Bradford assay and equal amounts of protein were added to a slurry of Dynabeads Protein G (Thermo Fisher Scientific) and p300 antibody (NM11, Santa Cruz Biotechnology). Samples were allowed to incubate 3 hr at 4°C before washing twice with Tris buffer + 0.01 Triton. Washed beads were then directly added to radioactive assays or run on gels as described previously. For radioactive assays, beads were incubated with 1 µM hot acetyl-CoA and 25 µM biotinylated peptide for 1 hr before quenching reaction in 2% TFA + 5% Streptavidin Sepharose High Performance (v/v, Cytiva).

## Acknowledgement

**Acknowledgements**

p300 plasmids were a kind gift from Dr. Daniel Panne, Leicester Institute of Structural and Chemical Biology, UK. Pan-butyryl antibody was the kind gift of Dr. Yingming Zhao, University of Chicago, USA. Recombinant SIRT2 was kindly provided by Dr. Mark Klein. H3 and H4 proteins and peptides were kindly provided by Dr. Vyacheslav Kuznetsov, Dr. Wallace Liu, and Lily Miller. The authors thank Dr. James Dowell and Eric Armstrong for their helpful discussions of mass spec analysis. The authors also thank Dr. Mark Klein, José Moran, and Dr. Spencer Haws for their helpful discussions of HAT assays. J.M.D. is co-founder of Galilei Biosciences and a member of the scientific advisory board for Evrys Bio.

## Additional information

### Funding

| Funder | Grant reference number | Author |
| --- | --- | --- |
| National Institutes of Health | GM059785 | John M Denu |
| National Science Foundation | GRFP | Sydney Thomas |

The funders had no role in study design, data collection and interpretation, or the decision to submit the work for publication.

### Author contributions

Sydney P Thomas, Conceptualization, Data curation, Formal analysis, Investigation, Methodology, Validation, Visualization, Writing – original draft, Writing – review and editing, Software; John M Denu, Conceptualization, Investigation, Methodology, Project administration, Supervision, Visualization, Writing – original draft, Writing – review and editing

### Author ORCIDs

Sydney P Thomas ⬤ http://orcid.org/0000-0001-8261-1195
John M Denu ⬤ http://orcid.org/0000-0001-9415-0365

### Decision letter and Author response

Decision letter https://doi.org/10.7554/eLife.72171.sa1
Author response https://doi.org/10.7554/eLife.72171.sa2

## Additional files

### Supplementary files

- Transparent reporting form
- Source data 1. Raw Data and Replicates for Enzyme Assays.
- Source data 2. Full Western Blots and Replicate Blots.

### Data availability

Mass spectrometry data has been uploaded to MassIVE (MSV000087800). All other source data is included in Source Data 1 and 2.

The following dataset was generated:

| Author(s) | Year | Dataset title | Dataset URL | Database and Identifier |
| --- | --- | --- | --- | --- |
| Denu J, Thomas S | 2021 | Data for: Short-chain fatty acids activate acetyltransferase p300/CBP | ftp://massive.ucsd.edu/MSV000087800/ | MassIVE, MSV000087800 |

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
