## [Editor Report]

This study investigates the mechanism of agents like butyrate as metabolites that affect histone acetylation. The authors make the unexpected and interesting finding that such metabolites can stimulate the activity of the acetyltransferase p300 rather than the commonly accepted concept that they block histone deacetylases. The authors show evidence that p300 stimulation involves acylation of Lys residues on its autoinhibitory loop. The authors have effectively responded to prior concerns raised by the reviewers. This study should be of broad interest to the epigenetic research community.

---

## [Decision Letter]

Thank you for submitting your article "Short-chain fatty acids activate acetyltransferase p300" for consideration by *eLife*. Your article has been reviewed by 3 peer reviewers, one of whom is a member of our Board of Reviewing Editors, and the evaluation has been overseen by Philip Cole as the Senior Editor. The following individual involved in review of your submission has agreed to reveal their identity: Saadi Khochbin (Reviewer #2).

Essential revisions:

Short-chain fatty acids (SCFAs), such as acetate, propionate, and butyrate, are long thought to function as inhibitors of HDACs. Here, Thomas and Denu challenge this textbook dogma by providing argument that activation of CBP/p300 catalytic activity following AIL domain acylation is a major contributor to histone hyperacetylation induced by SCFAs. In this study, the authors carry out proteomic and metabolic analyses to determine changes in histone modifications and metabolites in cells treated with SFCAs. The results show that propionate and butyrate, but not acetate, induce rapid and dose-dependent increases in histone acetylation. The propionate and butyrate are rapidly metabolized into acyl-CoAs (uM or sub-uM levels) in cells, while the cellular levels of these SCFAs remain at low concentrations (tens of μm levels) that are not able to inhibit HDACs. MS of histone PTMs suggest that the majority of new acetyl groups on acetylated histones are not generated from the C13-labeled SCFAs, suggesting that a different mechanism is at play. Interestingly, HAT assays of the nuclear extracts show propionyl- and butyryl-CoA induce does-dependent increases in the HAT activity, whereas free propionate and butyrate have no effect. in vitro HAT assays using recombinant p300 HAT show that p300 utilizes butyryl-CoA and propionyl-CoA for auto acylation, and the auto-acylation activates the enzyme of p300 HAT in a similar manner to its auto-acetylation by using acetyl-CoA. Finally, they show that inhibition of p300 enzymatic activity, but not inhibition of HDACs, reverses SFCA-induced histone hyperacetylation.

This paper reports very intriguing observations and proposes a novel but surprising idea, which in our views deserve large diffusion and should fuel new thoughts and constructive debates. SCFAs, especially butyrate, have been found to have multiple beneficial health effects in the body. Likely, the finding presented in this work will elicit more attention and research in this field to further investigate polypharmacological mechanisms of SCFAs in influencing physiology and pathology.

However, the manuscript also raises a number of questions that need to be carefully addressed. Particularly, the precise mechanisms that differentiate the action of propionate and butyrate from acetate on CBP/p300 activity remain obscure and the authors could strengthen their conclusions by some additional investigations. Acetyl-CoA is a much better cofactor than butyryl-CoA and propionyl-CoA for p300. It needs to be addressed why an acetate treatment, as opposed to propionate or butyrate treatment, does not lead to increases in CBP/p300 AIL acetylation and to an activation of these HATs. Also, this paper does not have sufficient data to suggest SFCAs do not function through HDAC inhibition in cells. The very low concentrations of propionyl-CoA and butyryl-CoA present in the cell and the only modest activation on p300 activity by the AIL acylation are also of concern.

1) A major puzzle is how the three SCFAs, acetate, propionate, and butyrate affect histone acetylation differently, under the proposed mechanism of p300 activation by autoacylation. In in vitro p300 HAT assay, acetyl-CoA exhibits the same effect as propionate and butyrate (Figure 3B, C and S4C-E) and adding acetate in medium increases acetyl-CoA levels in cells (Figure 2E and S2D). However, the increased acetyl-CoA levels fail to enhance p300 autoacetylation and activation. This discrepancy is confusing and should be resolved.

2) The authors argue that intracellular concentrations of propionate and butyrate are not high enough to inhibit HDAC in cells. However, this paper does not have sufficient data to show HDAC activities are not effectively inhibited by propionate and butyrate at low concentrations (e.g. 10 nM) in cells. Importantly, intracellular distribution of propionate and butyrate is heterogeneous. Key regions inside the cell could have disproportionally high levels of propionate and butyrate to effective block HDAC activity. The authors could use target validation techniques such as cellular thermal shift assay (CETSA) to validate or invalidate if butyrate binds HDACs in cells at low concentrations (e.g., under 1 mM treatment condition).

3) Butyryl-CoA is a very poor substrate of p300 compared with acetyl-CoA and propionyl-CoA, however butyrate leads to the strongest histone hyperacetylation. In addition, in cells butyryl-CoA is at only tens of nM levels compared to sub-uM of acetyl-CoA and propionyl-CoA (Figure 3). This discrepancy needs to be resolved. Figure 3A also needs to be carefully and quantitatively re-examined to substantiate if butyryl-CoA is a better substrate than acetyl-CoA and propionyl-CoA in p300 autoacylation.

4) The mechanism underlying the conversion of propionate and butyrate into their corresponding CoA derivatives described here as a key event in activating CBP/p300 is not clear. Published data indicate that ACSS2 is able to synthesize crotonyl-CoA (PMID 25818647). It is very likely that ACSS2 can also synthesize propionyl-CoA and butyryl-CoA, and if so, its down regulation should affect p300 auto-acylation (PMID 25818647). However, here the authors show that ACSS2 knock-down has no significant effect on propionate or butyrate-induced histone acetylation. Could this be due to inefficient knock-down of ACSS2? Experimental data are needed to address this discrepancy.

---

## [Author Response]

Essential revisions:Short-chain fatty acids (SCFAs), such as acetate, propionate, and butyrate, are long thought to function as inhibitors of HDACs. Here, Thomas and Denu challenge this textbook dogma by providing argument that activation of CBP/p300 catalytic activity following AIL domain acylation is a major contributor to histone hyperacetylation induced by SCFAs. In this study, the authors carry out proteomic and metabolic analyses to determine changes in histone modifications and metabolites in cells treated with SFCAs. The results show that propionate and butyrate, but not acetate, induce rapid and dose-dependent increases in histone acetylation. The propionate and butyrate are rapidly metabolized into acyl-CoAs (uM or sub-uM levels) in cells, while the cellular levels of these SCFAs remain at low concentrations (tens of μm levels) that are not able to inhibit HDACs. MS of histone PTMs suggest that the majority of new acetyl groups on acetylated histones are not generated from the C13-labeled SCFAs, suggesting that a different mechanism is at play. Interestingly, HAT assays of the nuclear extracts show propionyl- and butyryl-CoA induce does-dependent increases in the HAT activity, whereas free propionate and butyrate have no effect. in vitro HAT assays using recombinant p300 HAT show that p300 utilizes butyryl-CoA and propionyl-CoA for auto acylation, and the auto-acylation activates the enzyme of p300 HAT in a similar manner to its auto-acetylation by using acetyl-CoA. Finally, they show that inhibition of p300 enzymatic activity, but not inhibition of HDACs, reverses SFCA-induced histone hyperacetylation.This paper reports very intriguing observations and proposes a novel but surprising idea, which in our views deserve large diffusion and should fuel new thoughts and constructive debates. SCFAs, especially butyrate, have been found to have multiple beneficial health effects in the body. Likely, the finding presented in this work will elicit more attention and research in this field to further investigate polypharmacological mechanisms of SCFAs in influencing physiology and pathology.However, the manuscript also raises a number of questions that need to be carefully addressed. Particularly, the precise mechanisms that differentiate the action of propionate and butyrate from acetate on CBP/p300 activity remain obscure and the authors could strengthen their conclusions by some additional investigations. Acetyl-CoA is a much better cofactor than butyryl-CoA and propionyl-CoA for p300. It needs to be addressed why an acetate treatment, as opposed to propionate or butyrate treatment, does not lead to increases in CBP/p300 AIL acetylation and to an activation of these HATs. Also, this paper does not have sufficient data to suggest SFCAs do not function through HDAC inhibition in cells. The very low concentrations of propionyl-CoA and butyryl-CoA present in the cell and the only modest activation on p300 activity by the AIL acylation are also of concern.

We thank the reviewers for their insightful summary and excitement for this novel work.

1) A major puzzle is how the three SCFAs, acetate, propionate, and butyrate affect histone acetylation differently, under the proposed mechanism of p300 activation by autoacylation. In in vitro p300 HAT assay, acetyl-CoA exhibits the same effect as propionate and butyrate (Figure 3B, C and S4C-E) and adding acetate in medium increases acetyl-CoA levels in cells (Figure 2E and S2D). However, the increased acetyl-CoA levels fail to enhance p300 autoacetylation and activation. This discrepancy is confusing and should be resolved.

First, an important clarification: Acetate addition does not lead to substantial increases in acetyl-CoA, which is consistent with the lack of stimulated histone acetylation. Thus, we would argue that our proposed mechanism is well-aligned with the observations. As stated in lines 284-301, the central difference amongst the three SCFAs is that exogenously added acetate is not rapidly converted to acetyl-CoA. The reviewer is right to notice that total levels of acetyl-carnitine and acetyl-CoA do increase over time in Figures 2E and S2D; however, this occurs in every condition (including the untreated control), indicating that the increase in acetyl-CoA is due to factors other than exogenous addition of acetate. Instead, levels of labeled acetylCoA/carnitine in our acetate treated samples remains low, and total levels closely mirror those in untreated controls.

Importantly, these results indicate that in the cell lines we tested, the activity of ACSS2 (acetyl-CoA synthetase (nuclear-cytoplasmic)) is relatively low compared to the (unknown) synthetases that convert propionate and butyrate to their respective esterified CoA forms. This is another extremely critical point of our work and is further detailed in subsequent responses to reviewers.

2) The authors argue that intracellular concentrations of propionate and butyrate are not high enough to inhibit HDAC in cells. However, this paper does not have sufficient data to show HDAC activities are not effectively inhibited by propionate and butyrate at low concentrations (e.g. 10 nM) in cells. Importantly, intracellular distribution of propionate and butyrate is heterogeneous. Key regions inside the cell could have disproportionally high levels of propionate and butyrate to effective block HDAC activity. The authors could use target validation techniques such as cellular thermal shift assay (CETSA) to validate or invalidate if butyrate binds HDACs in cells at low concentrations (e.g., under 1 mM treatment condition).

We do not argue that HDAC inhibition does not occur during high physiological exposure to SCFAs, but that given the compelling data in this paper and with a more insightful review of prior studies, the contribution by HDAC inhibition could be much lower than previously thought. This case can be made best with low levels of propionate treatment, which on a molar-to-molar basis leads to similar hyperacetylation as butyrate, despite the fact that butyrate is a much better HDAC inhibitor than propionate. We fully understand that we are proposing a novel mechanism that appears to go against the prevailing theory. However, these mechanisms are not mutually exclusive. We argue that under non-supra physiological levels of SCFAs, p300/CBP HAT activation is a viable mechanism to explain histone hyperacetylation.

Regarding the possibility that different compartments of the cell might have “disproportionally high levels of propionate and butyrate to effective block HDAC activity”: this is certainly possible, but our studies are focused on nuclear and cytoplasmic events, where we have measured total SCFA levels. We are not aware of any studies that show free SCFA concentrations are different between nucleus and cytoplasm, and given that these compartments make up the majority of the cell volume, we believe our numbers are accurate reflections of SCFAs concentrations for the processes we investigated.

The reviewer suggested that we could perform “cellular thermal shift assay (CETSA) to validate or invalidate if butyrate binds HDACs in cells at low concentrations.” This is an interesting suggestion, but unfortunately interpreting such an experiment is not as straight forward as suggested. Since there is currently no consensus as to which specific HDACs are inhibited by butyrate, CETSA assays would likely prove inconclusive since thermal shifts for every HDAC would be difficult to report in a single assay. We believe our data measuring the apparent IC50 (inhibition) of SCFAs in nuclear extracts provides a more direct assessment of the binding to HDACs in a relevant context. To make this point more strongly, we have included the raw data in Supplemental Figure 1E. IC50 values are reported in Table 1.

3) Butyryl-CoA is a very poor substrate of p300 compared with acetyl-CoA and propionyl-CoA, however butyrate leads to the strongest histone hyperacetylation. In addition, in cells butyryl-CoA is at only tens of nM levels compared to sub-uM of acetyl-CoA and propionyl-CoA (Figure 3). This discrepancy needs to be resolved. Figure 3A also needs to be carefully and quantitatively re-examined to substantiate if butyryl-CoA is a better substrate than acetyl-CoA and propionyl-CoA in p300 autoacylation.

Because butyrate is the best HDAC inhibitor of the three, at higher levels (~1 mM) both HAT activation and HDAC inhibition could account for the more robust histone acetylation. However, at 100 µM, the hyperacetylation induced by propionate and butyrate are essentially identical (Figure 4). We have included a discussion of this point starting at line 352.

Our auto-acylation data in Figure 3B and 3C suggests that butyryl-CoA, propionyl-CoA acetyl-CoA and are equally effective substrates for autoacylation, at least under the conditions assayed. It is reported that butyrylCoA is a poor substrate for trans-butyrylation of histones (see Kaczmarska et al. and our data in Figure 1). However, our auto-acylation data suggests that the mechanisms for auto- and trans-acylation differ. A discussion of this point is included in lines 330-344. Our next major project is to perform a detailed mechanistic analysis of the acyl-specificity for auto-acylation, and to examine whether the resulting activity enzyme displays different specificities in trans-acetylation.

As to the concentration levels, the literature does suggest that acetyl-CoA is more abundant than propionyl- or butyryl-CoA under normal glucose-rich cellular conditions (see Simithy et al.). However, acetyl-CoA generation is tightly regulated with cell type and energy state, and the levels of short-chain acyl-CoAs will be dictated by the availability of extracellular SCFAs and other non-glucose metabolism, particularly amino acids and FAs. Thus, even a two-fold increase in propionyl- or butyryl-CoA could provide a new acyl-CoA pool that was previously unavailable to p300, allowing p300 to respond to specific increases in new acyl-CoA species. Another important consideration is the rate at which specific acylations are removed by deacetylases, potentially increasing the lifetime of p300 activation via certain auto-acylated forms. This will be a profoundly exciting possibility to explore.

4) The mechanism underlying the conversion of propionate and butyrate into their corresponding CoA derivatives described here as a key event in activating CBP/p300 is not clear. Published data indicate that ACSS2 is able to synthesize crotonyl-CoA (PMID 25818647). It is very likely that ACSS2 can also synthesize propionyl-CoA and butyryl-CoA, and if so, its down regulation should affect p300 auto-acylation (PMID 25818647). However, here the authors show that ACSS2 knock-down has no significant effect on propionate or butyrate-induced histone acetylation. Could this be due to inefficient knock-down of ACSS2? Experimental data are needed to address this discrepancy.

Although the exact mechanisms that convert propionate and butyrate to their corresponding acyl-CoAs are not yet clear (for an excellent review of current knowledge on the subject, see Trefely et al.), our data show that butyrate and propionate but not acetate are rapidly (minutes) converted to their acyl-CoAs and acyl-carnitines. While ACSS2 might appear to be a promising candidate for propionate and butyrate, our data and others suggest that ACSS2 is not likely the enzyme responsible for esterifying propionate or butyrate to CoA.

ACSS2 is highly specific for acetate. We refer this discussion to two important JBC papers, one from Brown and Goldstein (Luong et al, 2000) and the other from Fujino et al (2001). As substrates, acetate is >30-fold better than propionate, and no significant activity was measured with butyrate.

Because we observe no significant increase in acetyl-CoA or hyperacetylation by acetate addition to cells, we expect that ACSS2 expression and activity is extremely low in the cell lines analyzed here. This is very common under typical cell culture growth conditions.

Thus, the knockdown of ACSS2 was predicted to exhibit little difference with acetate. Importantly, the fact that ACSS2 knockdown also had no significant effect on propionate- and butyrate-induced acyl-CoA levels or histone hyperacylation supports the restricted substrate specificity of ACSS2 (Fujino et al, and Luong et al).

We would argue that a comprehensive understanding of SCFA conversion to acylCoAs is outside the scope of this paper.